# Integrative Approach of MAP and Active Antimicrobial Packaging for Prolonged Shelf-Life of Composite Bottle Gourd Milk Cake

**Rekha Chawla** [1,*], **Sivakumar Shanmugam** [1] , **Jasbir Singh Bedi** [2], **Selvamuthukumaran** [3], **Deep N. Yadav** [4] and **Rahul Anurag** [5]

1   Department of Dairy Technology, Guru Angad Dev Veterinary and Animal Sciences University, Ludhiana 141004, India
2   Centre of One Health, College of Veterinary Sciences, Guru Angad Dev Veterinary and Animal Sciences University, Ludhiana 141004, India
3   Department of Food Science and Technology, Hamelmalo Agricultural College, Hamelmalo 397, Eritrea
4   Technology Transfer Division, ICAR-Central Institute of Post-Harvest Engineering and Technology, Ludhiana 141004, India
5   Agriculture Structures and Environmental Control, ICAR-Central Institute of Post-Harvest Engineering and Technology, Ludhiana 141004, India
*   Correspondence: mails4drrekha@gmail.com or rekhachawla@gadvasu.in

**Abstract:** The current research explored the integrative effect of antimicrobial edible films and modified atmosphere packaging (MAP) on the quality parameters and shelf-life of bottle gourd burfi, which is a heat-desiccated composite Indian confection. The edible antimicrobial films prepared using a combination of nisin and natamycin (NANIF) were evaluated for their antimicrobial activity as the first line of defense against *Bacillus cereus* and *Aspergillus niger*. The product was wrapped in developed films, which was followed by flushing of the altered environment employing MAP in a closed PP box and evaluation during refrigerated storage at $4 \pm 2$ °C, comparing the product with the control counterpart. During this period, the physicochemical, sensory, and microbiological status of the product was assessed. Results indicated a significant ($p \leq 0.05$) variance between the two kinds of samples wherein the antimicrobial film produced excellent results in terms of being less supportive toward microbial growth, thereby extending the life of film-treated samples beyond 35 days compared to the control (21 days). In addition, the product conformed to the legal standards of microbiological count well under the permissible limits laid by the FSSAI. Furthermore, the sensory characteristics of the product did not change much, illustrating the significance of the integrative approach.

**Keywords:** bottle gourd burfi; active packaging; modified atmospheric packaging; antimicrobial edible films; sensory evaluation

## 1. Introduction

To meet the needs of contemporary consumers and to keep abreast with the latest technological demands, enhanced shelf-life has become the prerequisite amongst available choices. Therefore, the safety of food products has become the chief concern for the food industry owing to various rapid reactions occurring during storage, leading to undesirable changes in microbiological and physicochemical characteristics [1]. In addition, due to the emergence of foodborne pathogens and their role in human diseases, emphasis has been laid down on the advancements in active packaging, aimed at inhibiting the growth and multiplication of these pathogens. Such initiatives employing packaging tools have been significantly highlighted in the recent past [2]. Research has proven that regulating oxygen and keeping microorganisms at bay are the two primary measures to improve the quality and shelf-life of food products [3]. However, the pace of food deterioration can be impeded by covering the food products with a protective film/coating that acts as a sheath

to protect the food and thereby possibly enhance its quality and shelf-life [4]. However, the product's life is at stake without modern tools. Hence, over the last few years, packaging has evolved to incorporate various substances to provide different kinds of desirable effects in food products, some of which include coloring agents, antimicrobials, antioxidants, oxygen, moisture, and ethylene scavengers, whereas some foods require the emitters such as ethylene and carbon dioxide [5]. Amongst these, antimicrobials compounds impregnated in edible films have recently become the choice of researchers to bring the desirable preferences of extended shelf-life. Different raw materials of agricultural or marine origin such as polysaccharides, lipids, and/or proteins can be employed for producing biopolymer-based packaging films, which have been proven to be potential barriers to moisture, oxygen, and solute exchange and are an excellent biodegradable alternative to plastic films [6,7], with added antimicrobial compounds to provide additional protection against the proliferation of microorganisms. The upside of these natural antimicrobials includes various desirable qualities wherein bacteriocins offer an immense potential application in edible films such as inhibitory action against bacterial spoilage, natural origin, resistance to acids and elevated temperatures, etc. [8].

Among various bacteriocins, nisin is the most extensively studied FDA-approved, GRAS antimicrobial peptide generated by *Lactococcus lactis*, a lactic acid bacterium, and it is known for its broad applicability in active food packaging [9]. It exhibits intense inhibitory activity against an extensive spectrum of Gram-positive microbes, including major foodborne pathogens (*Listeria monocytogenes*, *Bacillus cereus*), and it is also effective against spore germination [10,11]. On the other hand, natamycin has aroused significant interest in the packaging industry for its effective antifungal action against various spoilage fungi and mods (*Aspergillus niger*, *Penicillum*) [12].

Furthermore, altering the microenvironment associated with the product within the packaging system is another approach toward adding hurdles to the proliferation of organisms and a way forward toward improving the shelf-life [3]. MAP in combination with low-temperature storage can significantly enhance the storage stability and quality preservation of a wide variety of products such as dairy, meat, fish, and poultry products [13,14]. However, very few instances of combined application of MAP with edible film could be traced in the literature, especially in the case of dairy products involving sweetmeats [13–16]. These delicacies not only have a strong impact on socioeconomic prospects but also possess a revealing impact on the nutrition and social well-being of people [17]. An average of 6.5% of total liquid milk produced is diverted for *Khoa* or condensed milk products [18]. The preparation of such products not only allows the consumption of milk-based products amongst various age groups but also increases nutritional value many times over due to the desiccation of contents and removal of water from the milk.

Under an array of choices, composite dairy products are becoming the healthful choice of a conscious society. These composite sweets not only taste delicious but are a reservoir of healthful components in terms of nutrients. In addition, the incorporation of commodities other than milk adds vital nutrients to the ultimate sweet, which are sometimes otherwise absent or available in trace amounts in the milk [19]. The addition of bottle gourd (Lagenaria siceraria) shreds to milk for the preparation of bottle gourd burfi is one of the relevant examples under the category of vegetable–dairy-based composite confections. It is an important vegetable of the family Cucurbitaceae and has its origin in Africa and Asia, especially India. Nutritionally, it is considered a reservoir of a myriad of benefits including diuretics apart from being loaded with B vitamins, Vitamin C, fiber, and minerals such as calcium, potassium, magnesium, phosphorus, etc. Scientific investigations on the preparation of this product are scant, and only a few investigations have been undertaken on its preparation methodology [20]. Furthermore, the product had been reported to have a shelf-life of only 12 days at room temperature wherein further scientific investigations toward extended life have not been undertaken. Taking these gaps further, an integrative approach was utilized to assess the impact of MAP in combination with

edible antimicrobial packaging for composite sweets with an aim to extend the shelf-life of this specific product.

## 2. Materials and Methods

### 2.1. Material

The basic biopolymer, i.e., corn starch was derived from Loba Chemie Pvt. Ltd., Mumbai, Maharashtra, while anhydrous glycerol was drawn from the Advent Chembio (Mumbai, India), -for edible film preparation. Analytical grade chemicals and distilled water were used for reagent preparation. The media for microbiological investigation were procured from Himedia Lab, Mumbai, India. Nisinproq was procured from Proquiga Biotech, La Coruna, Spain, whereas natamycin was received from Shandong Freda Biotechnology Co., Ltd., Shandong, China. For the preparation of control edible films, corn starch was employed involving casting solution procedure using starch and glycerol, and it was added at pre-standardized values [14]. The detailed procedure of edible film preparation could be fetched from [15] (Figure 1) with the only difference being the absence of antimicrobial agents.

Dissolution of starch (biopolymer) @5% in about 60% water

⬇

Homogenization with Ultraturrax homogenizer, @5000-5500 rpm for 10 min

⬇

Heating until gelatinization (65-70ºC) with continuous stirring @750rpm

⬇

Dissolve glycerol (@3%) in 22% water separately

⬇

Add glycerol solution to gelatinized mixture and stir @750rpm for about 10 min

⬇

Add respective sterilized antimicrobial solution @10% into the mixture*

⬇

Stirring the contents for 15 min

⬇

Cooling (45-50˚C)

⬇

Bubble free plating for solification in sterilized petri-plates (Polypropylene, 30 cm × 20 cm)

⬇

Drying (16 –18 h)

⬇

Storage until further use

**Figure 1.** Preparation of edible packaging film using casting method.

Preparation of Nisin Stock solutions: First, 2 g of nisin (potency $\geq$ 900 IU/mg) was dissolved in 9 mL 0.02N HCl to give a solution with potency equal to 200,000 IU/mL, and further dilutions consisting of 2.50 mL of nisin stock solution + 7.50 mL distilled water were mixed together to prepare 10 mL of solution, which was added to 100 mL of film solution to obtain the desired ratio of 5000 IU.

Similarly, to prepare Natamycin Stock solutions: First, 0.1 g of natamycin was dissolved in 100 mL of distilled water to give a solution with potency equal to 1000 μg/mL. The composite usage of both the antimicrobials was utilized involving the mixing of 2.5 mL nisin stock + 3 mL natamycin stock + 4.5 mL distilled water to obtain the desired above-mentioned concentrations of nisin and natamycin, respectively. The resulting antimicrobial solutions were sterilized using 0.22 micron PVDF hydrophilic membrane syringe-driven filters.

## 2.2. Fabrication and Assessment of Antimicrobial Films

To prepare the antimicrobial edible film, part of the water of the film solution was substituted with a similar volume of an antimicrobial solution comprising predetermined concentrations of the antimicrobials to produce films with final concentrations of 5000 IU of nisin/mL and 30 μg of natamycin/mL in solution amongst various tested concentrations (Figures 2 and 3). A detailed investigation regarding the MIC of natamycin and nisin has been published in the preceding study, illustrating the choice of final concentrations for the present investigation [15].

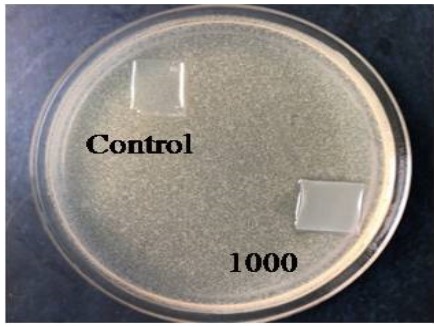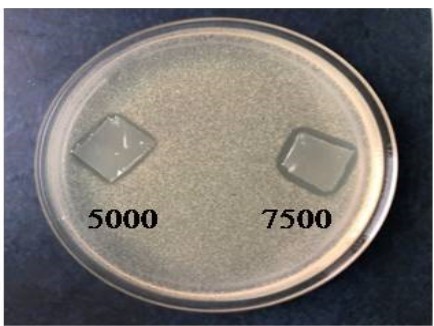

**Figure 2.** Antimicrobial activity of edible films incorporated with different concentrations (IU/mL) of nisin on the growth of *Bacillus cereus*.

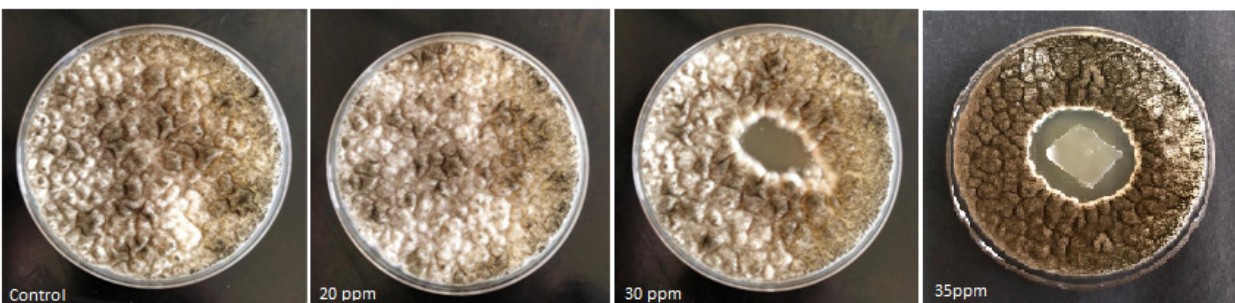

**Figure 3.** Antimicrobial activity of edible films incorporated with different concentrations (μg/mL) of natamycin on the growth of *Aspergillus niger*.

For full coverage of the product's surface using films, petri plates of size (150 mm with a height 19 mm) made of polypropylene (PP) were used during the casting procedure. To check the efficacy of controlled and prepared films, agar disk-diffusion assay was harnessed. The growth-limiting activity of the control and treated films was finally measured by calculating the total circumference of the zone (taking into account the film plus the inhibition zone) in a cross-section from various points (Figures 4 and 5).

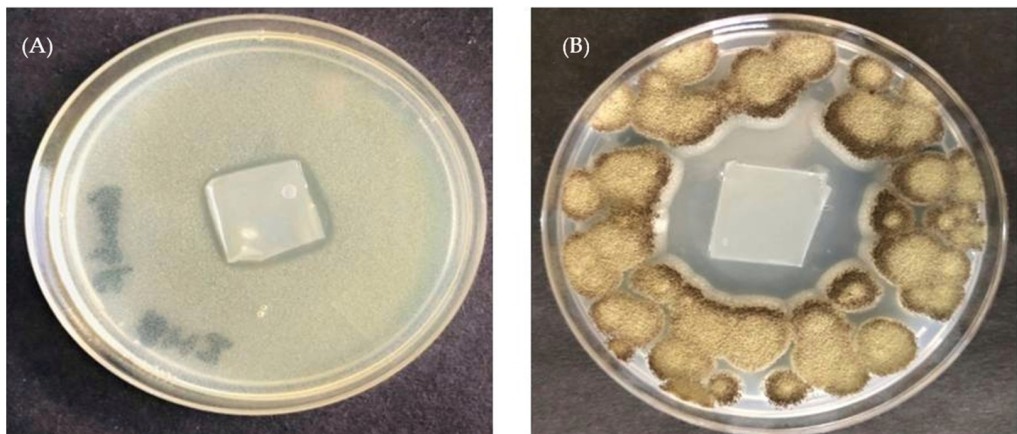

**Figure 4.** Antimicrobial activity of edible films (NANIF) on growth of (**A**) *Bacillus cereus* and (**B**) *Aspergillus niger*.

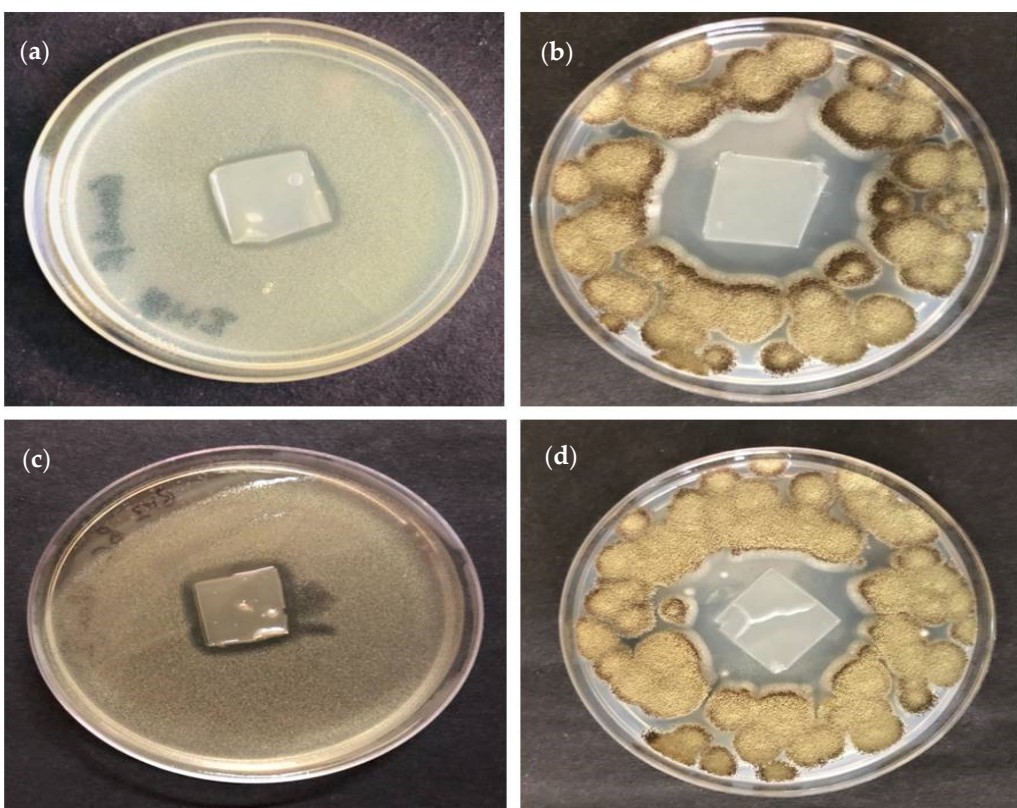

**Figure 5.** Antimicrobial activity of (**a**) filtered edible film natamycin and nisin-based film (NANIF) against *Bacillus cereus*, (**b**) filtered edible film (NANIF) against *Aspergillus niger*, (**c**) unfiltered edible film (NANIF) against *Bacillus cereus*, (**d**) unfiltered edible film (NANIF) against *Aspergillus niger*.

### 2.3. Residual Losses Estimation Study
Preparation of Unsterilized and Filter-Sterile Antimicrobial Films

To determine the effect of the filter sterilization of antimicrobial agents on the final concentration of prepared treated films, an antimicrobial assay of films containing unfiltered and filtered solutions of nisin and natamycin was conducted against *B. cereus* and *A. niger*. To prepare the same, predetermined effective concentrations of nisin (5000 IU/mL) and natamycin (30 µg/mL) were selected and nisin–natamycin edible film (NANIF) was prepared as referring to the above procedure outlined in Figure 1. Herein, the established quantity of both the antimicrobials was added as an exchange solvent with the conventional

water in a film-forming solution. However, due care was given to add an appropriate quantity of antimicrobials to produce films with 5000 IU nisin/mL and 30 μg natamycin/mL in the final solution. Thereafter, the nisin–natamycin solution was passed through a 0.22 μm filter-membrane before addition to the film solution in filtered antimicrobial films, whereas in the case of unfiltered antimicrobial films, the solution was added without employing membrane filtration to the resulting film solution.

### 2.4. Manufacturing, Wrapping, and Storage

The recipe demonstrated by Anurag and Chawla [20] was utilized for preparing composite milk-based sweet. After preparation, one set of samples was treated with NANIF films, and the product was packed in single units in 500 g thermoform boxes. However, the whole procedure was undertaken under almost aseptic conditions to lessen the likelihood of infection. The product was wrapped with utmost care, taking caution that all sides should be properly covered, leaving no space for the post-processing contamination (Figure 1). Both samples were additionally packed in containers made of thermoform material with size $18.5 \times 14.3 \times 4.8$ cm$^3$, which was manufactured from rigid coextruded PET sheet with a coating of coextruded LDPE film with the structure containing polyamide/polyethylene from Elixir Technologies, Bangalore, India. The bottle gourd burfi was packed in these boxes (capacity 500 g), following the flushing of the desired gas composition employing a machine supplied by PBI Dansensor and VAC Star-S 220 MP. To the boxes, a combination of $N_2$ and $CO_2$ was flushed in the ratio 70:30. The standard gas composition was identified from three different tested combinations of 50:50, 70:30, and 90:10 [14] based on its best support toward product acceptability. The product was refrigerated and evaluated for various parameters with a standard interval of one week each in the Food Processing Laboratory, CODST. The sample containers were randomly numbered and coded in a way to avoid biases and were stored in a refrigerator maintained at $4 \pm 2$ °C. The product was analyzed for its various parameters until considered unfit for human consumption. To produce error-free results, all samples were evaluated thrice.

### 2.5. Microbiological Investigation

Bacterial, mold and the presence of coliforms were investigated by the procedure demonstrated by Standard Methods for the Examination of Dairy Products [21].

### 2.6. Sensory Analysis

Considering an extended life of the composite product under MAP, a seven-day interval was followed to take up the product for sensory evaluation. At each interval, refrigerated samples were brought down to serving temperature and were presented to the faculty (semi-trained panelists) (15 in number) for further investigation and acceptance of the product. Samples were offered in pre-sterilized Petri plates and were evaluated, giving ample time to be judged accurately without introducing any means of bias. The quality assessment was based on a 9-point hedonic characterization [22], wherein different variables consisting of appearance, texture, sweetness, flavor, and overall acceptability were generated of parameters that could fully describe the composite sweetmeat. Furthermore, before the sensory evaluation, a round table discussion was also carried out to check the desirable attributes of the product, such as the fibrous appeal of the product, bottle green color, and an embedded nutty flavor.

### 2.7. Proximate Analysis and Degradative Products

The samples were checked for changes in major proximate attributes employing standard procedures illustrated by AOAC or IS:81, including moisture, protein, fat content, pH, titratable acidity, and total and reducing sugars, whereas storage samples were analyzed for tyrosine value, free fatty acids, hydroxyl methyl furfural (HMF) and thiobarbituric acid (TBA) value employing the relevant procedures.

*2.8. Statistical Investigation*

All the procedures were completed thrice to remove possible errors, and tests were used for suitable interpretation employing SPSS 16.0 software.

## 3. Results and Discussion

*3.1. Antimicrobial Activity of Treated Films*

GRAS-approved natamycin and nisin (NANIF) were used in combination to successfully prepare films using the bio polymer (starch) to check its efficacy against bacterial and fungal infections. These prepared films were assessed against Bacillus cereus and Aspergillus *niger* using the agar disk-diffusion technique. Based on the results of preliminary trials testing different nisin and natamycin concentrations in coupling, 30 μg/mL of natamycin and 5000 IU/mL of nisin were found to be the most effective concentrations for the best inhibitory results against both pathogens [23]. Clear zones around antimicrobial films were detected, indicating an affirmative antimicrobial action of both the added agents against the test organisms, whereas no zone was obtained around the control film (Figure 3). Results indicated that NANIF exhibited antimicrobial activity at tested concentrations, clearly portraying the dispersion of nisin and natamycin from the film to the surface of the set media, whereas the control film without any antimicrobial agent was unable to retard the microbial growth. Additionally, the presence of microbial growth even at the film–agar interface in the control film depicted that starch and glycerol exhibited no antimicrobial activity and rather acted as a substrate to promote the growth of microorganisms, being loaded with nutrients required for microbial proliferation. The antimicrobial assay depicted that the NANIF films produced an inhibition area of $6.01 \pm 0.29$ cm$^2$ and $16.95 \pm 1.08$ cm$^2$ against Bacillus *cereus* and Aspergillus *niger*, respectively (Figure 4).

The observations recorded in the antimicrobial assay indicated an affirmative interaction between the incorporated antimicrobials, as the presence of one did not alter the action or biological availability of the other. Studies on the same path employing antimicrobial agents such as nisin and natamycin in edible films prepared using different biopolymers have been conducted by various researchers. An examination by Ollé Resa et al. [24] showed the extremely effective antimicrobial action of edible starch films added with specified antimicrobials against both *L. innocua* and *S. cerevisiae* to improve the storage stability of Port Salut cheese. Likewise, the integration of nisin in the chitosan-based films showed quite effective antimicrobial activity against different genera of species, suggesting its potent role in preservation systems applicable to various food products [1]. Thus, on a similar platform, research undertaken by Pintado et al. [25], Ollé Resa et al. [26], and Berti et al. [27] reported the antimicrobial action of prepared films with both nisin and natamycin added.

*3.2. Depletion during Filtration*

Antimicrobial-embedded films prepared using a combination of nisin and natamycin (NANIF) were used for the evaluation of depletion studies. Results indicated that both filtered and unfiltered films exhibited antimicrobial action, clearly demonstrating the percolation of nisin and natamycin from the surface of the film to the media (agar) surface. The statistical investigation of the average values depicts that there was a non-significant difference among the two film samples in terms of inhibition zones against both the test organisms apart from certain marginal numerical differences. In the case of *B. cereus*, inhibition zones of 6.01 and 6.24 cm$^2$ were produced by filtered and unfiltered antimicrobial films, respectively, whereas the zone of inhibition of the two film samples fell in the range of 16.95–17.37 cm$^2$ when tested against *A. niger* (Figure 5). Therefore, the non-significant variance in the antimicrobial activity of the two film treatments may be attributed to the complete recovery of antimicrobial compounds (nisin and natamycin) through the membrane filter with proper sterilization without leaving any antimicrobial residue behind. Thus, it suggests that the concentration of the antimicrobial compound and its inhibitory activity remain unaltered after filter sterilization. Likewise, different antimicrobials other

than those used in the present study could also be explored for their efficacy against specific genera to reach a conclusion. However, in most of the literature surveys, predominantly, either the mentioned GRAS-approved antimicrobials or those derived from natural extracts such as essential oils or organic acids have been prevalently utilized.

*3.3. Progressive Microbiological Growth*

Microbiological contamination is a major limiting factor in extending the storage stability of dairy-based food products, causing untimely spoilage and leading to consumer unacceptability [28]. The current investigation aimed to explicate the synergistic effect of MAP and antimicrobial-incorporated edible films (NANIF) on the microbial quality and subsequent storage stability of bottle gourd burfi. Results indicated that there was a significant ($p = 0.000$) upsurge in the standard plate count (SPC) value with progressive storage in both samples, with the control samples displaying a significantly ($p = 0.000$) higher rate of microbial growth than antimicrobial film-treated (NANIF) samples. The standard plate count recorded was detected to be as no growth on day 0 and could be attributed to the high processing temperatures involved leading to heat desiccation of the product and the simultaneous eradication of the native microbiota. The initial nil SPC count also points out the nonappearance of any enduring heat-resistant spores of pathogenic or decaying bacteria in the prepared sweetmeat. However, the bacterial growth first emerged in control samples ($3.20 \pm 0.10$ log CFU/g) after one week, whereas the bacterial count could first be enumerated in NANIF samples ($3.46 \pm 0.09$ log CFU/g) after 14 days. During storage, the value increased to $4.31 \pm 0.02$ log CFU/g and $4.35 \pm 0.02$ log CFU/g in control and NANIF samples after 21 and 35 days of storage, respectively (Figure 6).

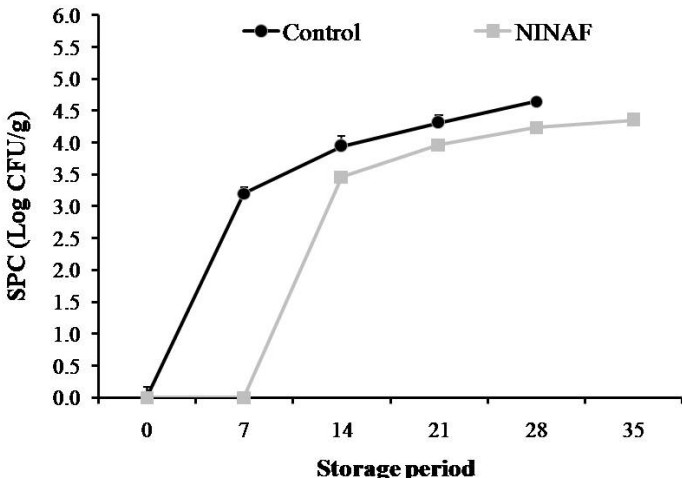

**Figure 6.** Effect of antimicrobial film along with MAP on standard plate count ($n = 3$) of composite sweetmeat during storage at 4 °C.

After 28 days of storage, the SPC count in control samples ($4.65 \pm 0.01$ log CFU/g) was quite close to the maximum permissible limit for khoa and khoa-based sweets as approved by FSSAI (4.87 log CFU/g) [28] along with bit disintegration of the fibrous cohesive structure of the product. Therefore, further analysis of the control samples after 21 days was discontinued, considering the shelf stability of the control product as 21 days. However, the analysis of NANIF samples was continued until the last day of estimated and planned storage, i.e., the 35th day. Even on this day, the count of the bacterial colonies fell in the range demonstrated by the Food Safety Standards Authority of India-approved permissible limits. At this point, the study was discontinued, keeping in view the loss of partial original flavor. The product till this stage had a cohesive close-knit texture and there was a possibility of keeping the product beyond this point as well without any microbial spoilage until 35 days, which otherwise has been marked as the shelf life of this product herein in this study.

In contrast, the coliform count remained nil in both the treatments throughout the storage period, thereby signifying the non-existence of any post-processing contamination and effective employment of utmost hygienic practices during preparation and packaging. The yeast and mold count also persisted at nil for both treatments during storage, except for their emergence in control samples ($0.36 \pm 0.06$ log CFU/g) only on the 28th day, suggesting the efficacy of incorporated natamycin against fungal proliferation. However, the packaged samples were free from any visible surface growth throughout the storage period. The lesser microbiological count in samples wrapped in treated film (NANIF) compared to the control product could be safely demonstrated as the potential effect of added antimicrobials, which were seeded in the primary film along with the altered environment in the form of a modified atmosphere, thereby ensuring better protection and preservation of product quality against both fungal and bacterial spoilage. These conclusions signify the harmonious effect of an altered environment and prepared antimicrobial films in enhancing the product life and storage stability by eradicating a crucial restraining factor, i.e., microbial contamination and subsequently improving the product's microbial quality.

The concurrent results are in correspondence to the observations made by Chawla et al. [29] in composite sweet *doda burfi*, wherein a gradual incremental pace was observed during the storage life of the product. In addition, Chawla et al. [30] reported a progressive growth count in a heat-desiccated milk product, khoa, during storage. However, the opposed results were obtained in our studies. This could be due to changes in the manufacturing procedure and the inherent nature of the raw material employed in composite sweetmeats as well as due to the repressive effect of MAP (altered gas environment) in the control. Supplementary to this, the MAP has also been promoted as a tool to enhance the life of parwal and *lal peda* in a study by Mishra et al. [31] and Jha et al. [32], respectively, where the authors suggested a reduction in fungal growth under low oxygen levels. Furthermore, Kalem et al. [33] also suggested reduced microbial growth in samples packed in bioactive edible films compared to control, signifying the role of bioactive edible films in greatly reducing the microbial count. They also indicated that the coliform count remained nil during storage owing to high-temperature heating as observed in the current study. Noor et al. [34] also suggested a similar increase in microbial growth in chevon sausages wrapped in calcium alginate edible films incorporated with *Asparagus racemosus* during storage.

### 3.4. Changes in Sensory Analysis

The sensory analysis of the product unleashed various expected and unexpected findings presented herein. Sensory evaluation of the bottle gourd burfi indicated a gradual but significant decline in acceptance in the overall liking of the product during storage, owing to the changes in its typical physical appearance (fibrous matrix with smooth body) and textural properties. The mean scores for different sensory properties evaluated have been depicted in Figure 7a–e. The sensory evaluation of control samples was conducted for 21 days considering the texture issues of the product, wherein the product could not hold its integrity for the full cut and lost cohesion between the fibrous particles. In contrast, the NANIF-treated samples were evaluated for their sensory attributes until the last day of planned storage, wherein the bacterial flora was found to be under permissible limits even on the 35th day. Nonetheless, various factors such as the loss of moisture and freshness, microbial contamination, proteolysis, lipolysis, etc., greatly influence the overall product acceptability, as it can lead to the development of undesirable flavor changes during product storage, thereby causing a falling of sensory scores [29]. The fall in these scores was significantly ($p = 4.4 \times 10^{-15}$) pronounced and projected in control contrast to the treated film samples (NANIF). It is worth mentioning herein that the typical decline in sensory scores recorded on the final day of antimicrobial film-wrapped samples was still on the lower side in comparison to the scores of the control even completing the doubled tenure of the life during storage. The body and texture scores are the most critical factors in the acceptance of this product as a fibrous structure bound in a cohesive network, and it

decreased from an initial score of 8.17 ± 0.17 to 7.17 ± 0.07 and 7.30 ± 0.06 in the control and NANIF samples after 21 and 35 days, respectively (Figure 7a). The decline in scores was significant ($p = 2.345 \times 10^{-14}$) between the storage days and among both the treatments applied. This drop in scores might be caused by a drop in moisture content and drying, leading to surface hardening of the product during storing at chill temperature and loss of gumminess during storage. The significant variance in scores among the treatments could be attributed to the antimicrobial properties of the NANIF films that protected the product from microbial degradation [33]. An analogous trend was observed in color and appearance scores that declined significantly ($p = 0.036$) during storage but with a small difference among treatments (Figure 7b). A similar decline in color, body and texture scores has been reported by Mishra et al. [31] in MAP-packed *parwal* sweet, Chowdhury et al. [35] in khoa, Chawla et al. [36] in doda burfi and Mahalingaiah et al. [37] in *kunda*. Likewise, chevon chunks wrapped in bioactive edible films (composite starch–chitosan impregnated with nisin and cinnamaldehyde) scored significantly higher texture scores compared to unwrapped control samples [38].

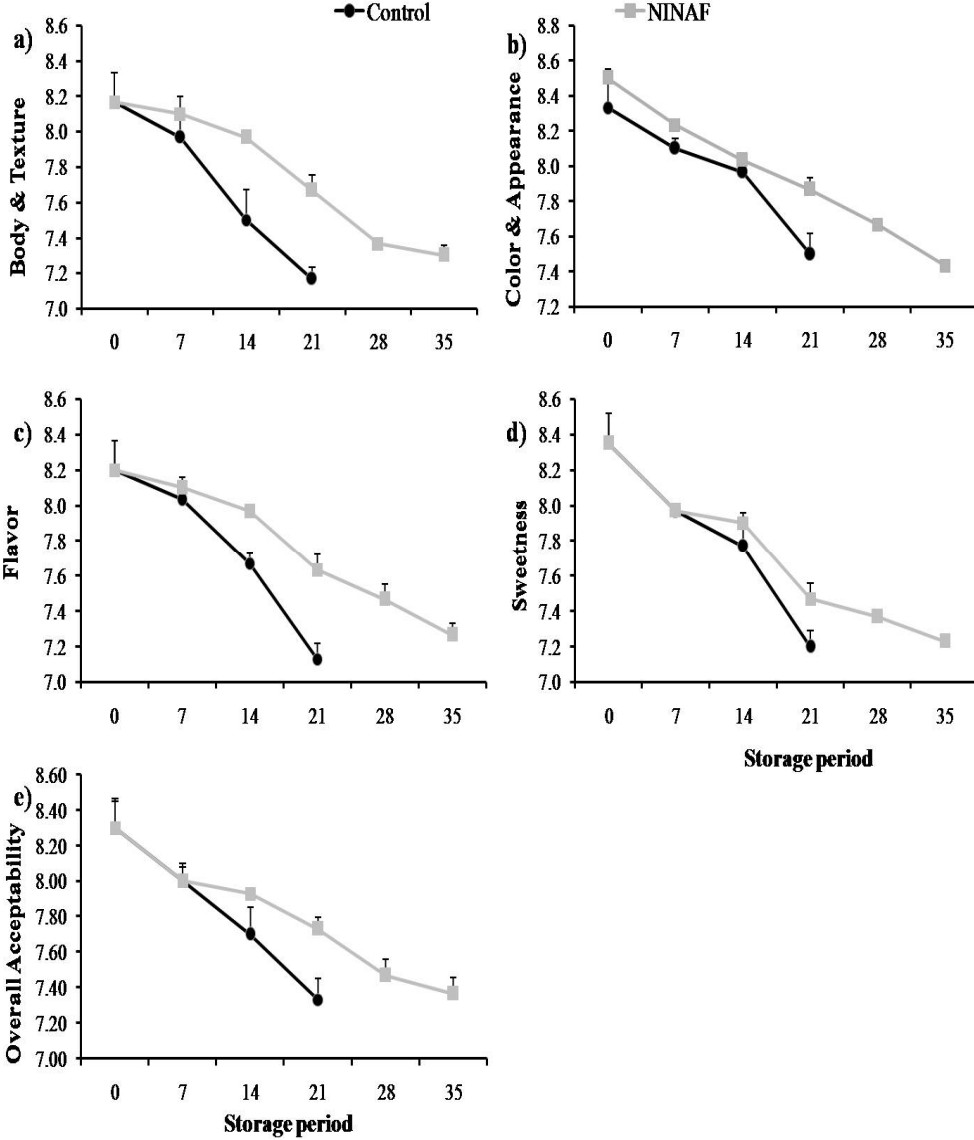

**Figure 7.** Effect of antimicrobial film along with MAP on (**a**) Body and texture, (**b**) Color and appearance, (**c**) Flavor, (**d**) Sweetness, and (**e**) Overall acceptability of composite sweetmeat (*n* = 9) during storage at 4 °C.

With progressive storage, the mean scores for flavor and sweetness also exhibited a gradual yet significant downward drift in both samples, with the drop in scores being statistically more prominent in control samples. The flavor scores witnessed a drop from an initial value of $8.2 \pm 0.15$ to $7.13 \pm 0.09$ and $7.27 \pm 0.07$ in control and NANIF samples on the 21st and 35th day of storage, respectively (Figure 7c). The decline in flavor scores might be due to the loss of moisture along with the chemical as well as microbial changes occurring in the product during storage, however, no off-flavor was perceived by the panelists despite the decline in scores. Similarly, the sweetness scores declined from an initial value of $8.35 \pm 0.07$ to $7.2 \pm 0.06$ and $7.23 \pm 0.03$ in the control and NANIF samples after 21 and 35 days, respectively (Figure 7d); however, the variation between both the samples was observed to be insignificant. These observations are in agreement with the results reported by Chawla et al. [29] in *doda burfi* and in MAP-packed khoa [35]. On similar grounds, Londhe et al. [39] and Jain et al. [3] reported similar trends in the hedonic characterization of stored brown peda and MAP-packed kalakand, respectively. In addition, Mushtaq et al. [40] suggested the use of a bioactive edible film for improving the sensory characteristics of the fresh Himalayan cheese.

The overall acceptability of the bottle gourd burfi declined significantly ($p = 0.0031$) but with a low pace during the progressive storage period for NANIF samples compared to the decline in scores being more significant and speedier in control. The initial average score for the overall acceptability of both samples was $8.3 \pm 0.10$, which declined to a concluding value of $7.33 \pm 0.12$ and $7.37 \pm 0.09$ in the control and NANIF product on the 21st and 35th day of storage, respectively (Figure 7e). The overall deterioration from the values obviously demonstrated and encourages the use of antimicrobial films in conjugation with MAP for products with similar characteristics. The decline in the overall acceptability could be related to the drying and loss of moisture and freshness along with the microbiological and chemical changes. In addition, the variation in scores among the two samples could be simply referred to the better protection and safeguarding of the product quality by the antimicrobial agents incorporated in NANIF films. Analogous results have been reported by different researchers, such as Chawla et al. [29], wherein the researchers proposed a diminishing trend in the sensory characterization of *doda burfi* while the product was in storage under refrigerated conditions. Chowdhury et al. [35] also reported a similar declining trend in overall acceptability during storage owing to the textural and flavor changes in the product. Likewise, Kalem et al. [33] and Noor et al. [34] also reported significantly higher sensory scores for cheese and meat samples packed in bioactive edible films compared to control.

### 3.5. Physiochemical Changes during Storage

Analyzing different physiochemical changes simply illustrates the rate of deterioration and changes in the quality of the product during storage. Various physiochemical attributes such as pH, titratable acidity, moisture, sugars, tyrosine, FFA, HMF, and TBA values were assessed at 7-day intervals to record the changes in the chemical nature of the product.

The stability of the product majorly relies on its moisture content, which also controls various other physiochemical properties indirectly. Both under and over-required moisture hampers with the desired quality characteristics of this delectable product, and it displayed a gradual yet significant decline in both the samples during storage. However, a significant ($p = 0.006$) difference between the two samples was observed only after 21 days of storage with the decrease being rapid in control samples. The highest mean moisture content was recorded as $24.95 \pm 0.22\%$ on day '0' of storage, which decreased to $21.40 \pm 0.12\%$ and $20.90 \pm 0.10\%$ in the control and NANIF samples after 21 and 35 days, respectively, with the decline being numerically more gradual in NANIF samples (Table 1). The overall moisture loss might be caused by the low temperature drying and surface evaporation occurring in closed packages during product storage [5,37]. The loss of moisture has also been predicted to cause another related defect of losing the cohesive structure of the product during storage. A similar decrease in moisture content during storage has

been reported by various researchers explaining the role of MAP and antimicrobial films. Chowdhury et al. [35] and Ghayal et al. [41] reported analogous trends in the moisture content of MAP-packed khoa and dietetic rabri while acknowledging the role of air composition in cutting water losses. Analogous observations were also recorded by Sutariya and Rao [4] for thirattupal as well as Jain et al. [3] in Kalakand and Chawla et al. [36] for *doda burfi*. These authors inferred a relation between the losses incurred during progressive storage with different factors such as storage temperature and the type of packaging material employed. Additionally, the relation between progressive moisture loss and the subsequent increase in the mechanical and sensory textural stiffness of the product during storage was also established by Chawla et al. [29].

**Table 1.** Physio-chemical attributes of bottle gourd *burfi* during storage at 4°C.

| Attributes | Storage Period (Days) | | | | | |
|---|---|---|---|---|---|---|
| | 0 | 7 | 14 | 21 | 28 | 35 |
| **pH** | | | | | | |
| Control | 6.603 ± 0.009 [Aa] | 6.563 ± 0.009 [Ab] | 6.507 ± 0.009 [Ac] | 6.310 ± 0.012 [Bd] | - | - |
| NANIF | 6.603 ± 0.009 [Aa] | 6.573 ± 0.012 [Aab] | 6.533 ± 0.009 [Ab] | 6.400 ± 0.015 [Ac] | 6.273 ± 0.020 [Ad] | 6.093 ± 0.022 [Ae] |
| **Titratable Acidity (%LA)** | | | | | | |
| Control | 0.084 ± 0.003 [Ac] | 0.238 ± 0.030 [Ab] | 0.293 ± 0.018 [Ab] | 0.380 ± 0.022 [Aa] | - | - |
| NANIF | 0.084 ± 0.003 [Ac] | 0.223 ± 0.026 [Ab] | 0.263 ± 0.016 [Ab] | 0.283 ± 0.033 [Ab] | 0.318 ± 0.048 [Aab] | 0.383 ± 0.027 [Aa] |
| **Moisture (%)** | | | | | | |
| Control | 24.95 ± 0.22 [Aa] | 24.06 ± 0.18 [Ab] | 22.97 ± 0.26 [Ac] | 21.40 ± 0.12 [Bd] | - | - |
| NANIF | 24.95 ± 0.22 [Aa] | 24.35 ± 0.27 [Ab] | 23.53 ± 0.15 [Ac] | 22.09 ± 0.10 [Ad] | 21.52 ± 0.22 [Ad] | 20.90 ± 0.10 [Ae] |
| **Ash (%)** | | | | | | |
| Control | 1.24 ± 0.01 [Ab] | 1.41 ± 0.05 [Ab] | 1.54 ± 0.02 [Aa] | 1.62 ± 0.01 [Aa] | - | - |
| NANIF | 1.24 ± 0.01 [Ab] | 1.26 ± 0.02 [Ab] | 1.32 ± 0.01 [Bb] | 1.33 ± 0.01 [Bb] | 1.56 ± 0.02 [Aa] | 1.61 ± 0.01 [Aa] |
| **Total sugars (%)** | | | | | | |
| Control | 26.35 ± 0.13 [Ac] | 27.40 ± 0.26 [Ab] | 27.90 ± 0.42 [Ab] | 29.23 ± 0.42 [Aa] | - | - |
| NANIF | 26.35 ± 0.13 [Ae] | 26.56 ± 0.26 [Ade] | 26.98 ± 0.11 [Ad] | 27.70 ± 0.08 [Bc] | 28.67 ± 0.22 [Ab] | 30.14 ± 0.21 [Aa] |
| **Reducing sugars (%)** | | | | | | |
| Control | 13.13 ± 0.40 [Ac] | 13.98 ± 0.29 [Ac] | 15.29 ± 0.19 [Ab] | 17.29 ± 0.31 [Aa] | - | - |
| NANIF | 13.13 ± 0.40 [Ad] | 13.46 ± 0.21 [Acd] | 14.21 ± 0.15 [Bc] | 15.72 ± 0.28 [Bb] | 16.63 ± 0.25 [Ab] | 17.89 ± 0.55 [Aa] |
| **Tyrosine value (mg/100g)** | | | | | | |
| Control | 0.056 ± 0.004 [Ad] | 0.113 ± 0.001 [Ac] | 0.123 ± 0.002 [Ab] | 0.136 ± 0.002 [Aa] | - | - |
| NANIF | 0.056 ± 0.004 [Af] | 0.098 ± 0.001 [Be] | 0.106 ± 0.002 [Bd] | 0.115 ± 0.002 [Bc] | 0.126 ± 0.002 [Ab] | 0.133 ± 0.002 [Aa] |
| **Free Fatty Acids (FFA) value (μeq/g)** | | | | | | |
| Control | 0.265 ± 0.021 [Ac] | 0.504 ± 0.079 [Ab] | 0.548 ± 0.076 [Ab] | 0.878 ± 0.090 [Aa] | - | - |
| NANIF | 0.265 ± 0.021 [Ad] | 0.298 ± 0.083 [Ad] | 0.468 ± 0.048 [Ac] | 0.735 ± 0.046 [Ab] | 0.828 ± 0.079 [Aab] | 0.982 ± 0.003 [Aa] |
| **Hydroxyl methyl furfural (HMF) value (μmoles/100g)** | | | | | | |
| Control | 10.13 ± 0.19 [Ad] | 13.34 ± 0.23 [Ac] | 14.68 ± 0.13 [Ab] | 15.96 ± 0.30 [Aa] | - | - |
| NANIF | 10.13 ± 0.19 [Af] | 12.94 ± 0.12 [Ae] | 13.70 ± 0.13 [Bd] | 14.39 ± 0.17 [Bc] | 15.04 ± 0.12 [Ab] | 16.38 ± 0.20 [Aa] |
| **Thio-barbituric acid (TBA) value (Absorbance)** | | | | | | |
| Control | 0.062 ± 0.005 [Ac] | 0.114 ± 0.004 [Ab] | 0.170 ± 0.009 [Aa] | 0.187 ± 0.005 [Aa] | - | - |
| NANIF | 0.062 ± 0.005 [Ae] | 0.092 ± 0.008 [Bd] | 0.123 ± 0.005 [Bc] | 0.146 ± 0.007 [Bb] | 0.155 ± 0.005 [Ab] | 0.186 ± 0.005 [Aa] |

Values are mean ± SE. Different letters (a–f) in same row denote a significant ($p \leq 0.05$) difference for storage period among days. Different letters (A,B) in same column denote a significant ($p \leq 0.05$) difference between control and NANIF sample.

Furthermore, the samples exhibited a slow yet statistically significant ($p = 0.000$) decline in pH when kept under storage with an initial pH value of 6.603 ± 0.009 for both samples. The pH of the control samples further declined to 6.310 ± 0.012 after 2 weeks, while the NANIF samples displayed a comparatively more gradual decline in pH, attaining the concluding value of 6.093 ± 0.022 after 35 days (Table 1). Furthermore, the pH varied significantly ($p = 0.06$) between different intervals during storage for both samples, where the reduction in pH of the control became statistically greater than NANIF-treated samples only on the 21st day while remaining only numerically higher during previous intervals. A similar decline in the pH was observed in milk cake packed in edible packaging during

storage [14]. For both the samples, this decline in pH may be associated with the growth of microorganisms in sweetmeat, which resulted in more acidic conditions or lower pH. Similarly, the titratable acidity of the samples depicted an overall increasing trend with a decline in pH at 4 °C. Generally, the increase in titratable acidity is directly linked with the microbial load of the stored product; i.e., an advanced bacteriological count results in higher titratable acidity, thereby indicating poor product quality. Initially, all samples exhibited the lowest titratable acidity, i.e., $0.084 \pm 0.003\%$ LA on day 0, which further continued to rise and reached a final value of $0.380 \pm 0.022\%$ LA and $0.383 \pm 0.027\%$ LA in control and NANIF-treated samples after 21 and 35 days of storage, respectively (Table 1). Regarding statistical analysis of the average values, the titratable acidity of both control and NANIF samples increased significantly throughout the storage period, whereas an insignificant variation ($p > 0.05$) was observed between the samples during the storage study. During storage, the gradual release of different acids ($CH_3COOH$, $C_3H_6O_3$, $HCOOH$, as well as other organic and FFA, amino acids, carbonic acids) as a result of some proteolytic and lipolytic reactions might be responsible for the overall decline in pH and subsequent increase in titratable acidity during storage. Furthermore, this might also be attributed to increased microbial growth, which has been proven responsible for the organic acids production during fermentation under anaerobic situations, which is maintained to slow down the growth of aerobic organisms during an altered environment, i.e., MAP [4,37,42]. The sluggish pace in augmented acidity for treated film samples may be attributed to the bacteriostatic/fungistatic or bactericidal/fungicidal action of the films, which had possibly deferred the microbial growth, thereby resulting in the lowered amount of acid produced. Similar observations have been recorded in various food products, such as brown peda, edible film-packed lor cheese, MAP-packed khoa, wrapped thirattupal, and MAP-packed cheese samples by Londhe et al. [39], Kavas et al. [43], Chowdhury et al. [35], Sutariya and Rao [4], and Barukčić et al. [44].

The ash content of bottle gourd burfi samples was analyzed to evaluate the variations in its mineral content during storage. Statistically, the ash content of both samples exhibited an overall statistically proven increase throughout the storage ($p \leq 0.05$) between the storage intervals and kinds of samples; with the increase being more pronounced in control samples. The ash content was recorded as $1.24 \pm 0.01\%$ for both the samples on the first day of storage, which eventually declined to $1.62 \pm 0.01$ and $1.61 \pm 0.01\%$ for both control and treated samples after 21 and 35 days, respectively, at refrigeration temperatures (Table 1). A similar upward trend in ash content was recorded by Choudhary et al. [42], and Gaikwad and Hembade [45] in buffalo milk concentrate (khoa) and buffalo milk ujani basundi.

During refrigerated storage, both total and reducing sugars displayed a significant ($p = 0.008$; $p = 0.013$, respectively) incline in both the samples. For total sugars, the initial mean value was recorded as $26.35 \pm 0.13\%$ for both samples on day '0', which significantly increased to a concluding mean value of $29.23 \pm 0.42\%$ & $30.14 \pm 0.21\%$ in both kinds on the 21st and 35th day of storage, respectively. Likewise, reducing sugars also increased from an original mean value of $13.13 \pm 0.40\%$ in both samples to $17.29 \pm 0.31\%$ in control after 21 days, while it elevated to a final mean value $17.89 \pm 0.55\%$ in NANIF-treated samples on the last day of storage ('35' day) (Table 1). The statistical analysis brought out a remarkable ($p \leq 0.05$) variation between the two kinds of samples with respect to their increase in both sugars. The overall increment in total and reducing sugars may be ascribed to the breakdown of compound starches to simpler sugars and further cessation to other simpler sugars in the product during storage. However, a comparatively greater increase in control samples might be caused by the microbial growth, which further boosted the rate of sugar breakdown, signifying the active role of antimicrobial NANIF films in maintaining lower values. An analogous increase in the quantity of reducing sugars of the doda burfi samples during storage at 4 °C was indicated by Chawla et al. [36]. A similar increase in the percent sugars of both types has been reported in milk cake prepared from carrots by Bajwa and Gupta [46] and in carrot candy prepared from honey by Durrani et al. [47].

Furthermore, the bottle gourd burfi was also checked and evaluated for its original fat and protein content as well as for its breakdown counterparts, i.e., free fatty acids (FFA) and tyrosine value (TV) on succeeding storage intervals to detect the lipolytic and proteolytic reactions and to elucidate the consequential variations in the fatty acid profile and protein composition of the product during storage. The initial fat and protein composition of the product was recorded as 21.73 ± 0.14% and 15.93 ± 0.33%, respectively, on day 0. The breakdown of fats in free fatty acids was clearly observable to a significant proportion ($p \leq 0.05$ $p = 0.000$) in all samples representing active lipolytic action. However, the variance in the values was statistically insignificant ($p > 0.05$) between the two kinds of treatment with the FFA value, with control samples being numerically higher than that of the treated samples. In control samples, the FFA value displayed an elevation from an initial value of 0.265 ± 0.021 μeq/g to 0.878 ± 0.090 μeq/g after 21 days, whereas it escalated to a final value of 0.982 ± 0.003 μeq/g in NANIF samples after 35-day storage at 4 °C (Table 1). The presence of high moisture content and free fats released during sweetmeat preparation along with lipase production due to microbial contamination might be responsible for the increased lipolytic activity, causing an upsurge in FFA value during storage. However, the removal of oxygen from the product packaging by MAP could be accountable for the increased life of the product, wherein oxygen was made a major limiting factor involved in the oxidation of lipids.

Likewise, a statistically significant ($p = 0.009$) upsurge in tyrosine value was recorded with advanced storage along with a substantial disparity amongst both the samples, where the rise in tyrosine value was statistically ($p = 0.000$) steadier in treated samples. From an original value of 0.056 ± 0.004 mg/100 g in both samples on the first day, the values rose to 0.136 ± 0.002 and 0.133 ± 0.002 mg/100 g after 21 and 35 days of refrigerated storage, respectively (Table 1). This could be referred to as the active mechanism of proteases in protein breakdown, which are released as a result of microbial contamination of the product under investigation, thereby suggesting that the treated film acted as a barrier to microbial growth, causing a more measured increment in NANIF samples compared to control. A similar surge in FFA and tyrosine was reported by Aggarwal et al. [30], Chawla et al. [36], and Choudhary et al. [42] in bottle gourd burfi, *doda burfi*, and khoa. Likewise, Jain et al. [3], Jha et al. [32], Mishra et al. [31], and Ghayal et al. [41] also reported delayed lipolytic activity in kalakand, lal peda, parwal and dietetic rabri during storage when packed under MAP. Analogous interpretations in terms of tyrosine value were drawn by Karunamay at al. [48], who highlighted the positive role of bioactive films in limiting the proteolytic reactions in paneer compared to control.

The samples were further analyzed for changes in their HMF and TBA content during refrigerated storage. The blanching treatment opted for bottle gourd burfi was a step toward the prevention of browning during the storage period. However, the presence of enzymes from the aerial flora was another obstacle to counter the efficacy of blanching. The lowest HMF content was recorded on the first day of storage (10.13 ± 0.19 μmoles/100 g) for both the samples, which shoot to 15.96 ± 0.30 in control samples only after 21 days, while NANIF samples exhibited a final HMF value of 16.38 ± 0.20 μmoles/100 g on the last day of refrigerated storage (Table 1). There was a significant ($p = 0.003$) increase in the HMF value of both samples with an increasing storage period. Likewise, the variation in HMF value among the two samples also remained statistically significant with the HMF value being higher in control samples, thereby signifying the higher susceptibility of the control samples to the occurrence of Maillard reaction. Similarly, the TBA values exhibited an upward drift while in storage, starting from an initial value (day 0) of 0.062 ± 0.005 in both samples and reaching the final values of 0.187 ± 0.005 and 0.186 ± 0.005 in the control and NANIF samples on the 21st and 35th day of refrigerated storage, respectively (Table 1). There was a noteworthy ($p \leq 0.05$) increase in TBA content throughout the storage period for both the control and NANIF-treated samples, with higher TBA values recorded in the control. The inclined TBA values might be accredited to the increase in lipid oxidation and hydrolysis in the product (bottle gourd burfi) during storage, which may have been

further accelerated by the increased microbial load in control compared to the antimicrobial film-treated samples. Analogous conclusions have been drawn by Ghayal et al. [41] and Mishra et al. [31] in dietetic rabri and parwal packed under modified atmosphere, with the lowest increase for both parameters observed in MAP gas compositions of 100% $N_2$ and 70% $N_2$:30% $CO_2$, respectively. A similar incline in HMF and TBA values during storage has also been reported by Jha et al. [32], Jain et al. [3], and Anurag et al. [19] in MAP-packed lal peda, MAP-packed kalakand, and bottle gourd burfi. In addition, Karunamay et al. [48] indicated a positive role of bioactive films in impeding the lipid oxidation reactions in paneer, thereby resulting in a lower TBA content in antimicrobial film-treated samples. Moreover, a comparable rise in the TBA content of queso blanco cheese wrapped in foxtail millet starch films has been reported by Yang et al. [49].

## 4. Conclusions

The current research corroborated the interactive action between MAP and antimicrobial films (NANIF) in terms of quality preservation and shelf-life improvement of the tested composite sweetmeat. The results of the present study depicted prolonged shelf-life beyond 35 days for samples with an antimicrobial primary layer compared to a life of 3 weeks in conventional samples kept at $4 \pm 2$ °C. The treated samples did not exhibit any symbolic spoilage parameter (surface growth and/or foul smell) during the entire storage and the same has been supported by the obtained sensory scores and microbiological count wherein the microbial load was found to fall within the allowable limits approved by FSSAI for heat-desiccated dairy products. In contrast to this, conventional films could not demonstrate any inherent antimicrobial activity and were ineffective in supporting prolonged shelf-life. The results suggested that the microbial count in control samples did not exceed the acceptable limit; however, they lost their desired structural integrity, which is the critical factor in product presentation. In addition, the treated samples demonstrated a statistically significant lower change rate in physiochemical and microbiological properties while their sensory scores remained noticeably higher compared to the control. Overall, the study ascertained the successful application of antimicrobial edible films in combination with MAP for effective quality maintenance and a significant increase in the shelf-life of the composite sweet.

**Author Contributions:** Conceptualization, R.C.; methodology, R.C.; software, J.S.B.; validation, Selvamuthukumaran, D.N.Y. and R.A.; formal analysis, R.A. and R.C.; investigation, R.C. and S.S.; resources, J.S.B., Rahul Anurag; data curation, D.N.Y.; writing—R.C.; writing—review and editing, S.; visualization, J.S.B.; supervision, R.C. and S.S.; project administration, R.C.; funding acquisition, R.C. All authors have read and agreed to the published version of the manuscript.

**Funding:** This work was financially aided by Department of Science and Technology under the project scheme-SYST (DST-SP/YO/080/2017).

**Institutional Review Board Statement:** Not applicable.

**Informed Consent Statement:** Not applicable.

**Data Availability Statement:** All the relevant data has been depicted in the form of figures and tables.

**Conflicts of Interest:** The authors declare no conflict of interest.

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
