# Peer review of "Integrative Approach of MAP and Active Antimicrobial Packaging for Prolonged Shelf-Life of Composite Bottle Gourd Milk Cake"

_coatings, doi:10.3390/coatings12081204_

Round 1

Reviewer 1 Report

In this manuscript, Chawla et al. describe the making and advantages of edible antimicrobial films containing nisin and natamycin and modified atmosphere packing on the quality and shelf-life of bottle gourd burfi. The authors show that their packaging method results in a much longer shelf-life from 21 days to greater than 35 days together while retaining sensory, taste, and texture. This work is interesting and I recommend publication in coatings, once the following comments are addressed:

  • Throughout the manuscript, the authors mention countless times p-values < or ≤ 0.05. However, the authors should report exact p-values.
  • On lines 439-440, the authors wrote: whereas an insignificant variation (p ≤0.05). Why is a p-value of ≤0.05 insignificant? Should it not rather be p >0.05?

Author Response

First and foremost, we are thankful for the reviewer for critical review of the manuscript so that gaps could be filled meticulously.

  1. As directed by the esteemed reviewer, wherever possible p values have been supplemented rather than broadly stating the level of significance, and at a few places level has also been deleted, considering an insignificant contribution to the overall interpretation of the results. A meaningful contribution has been done in revision.
  2. We are highly thankful to the reviewer for pointing out an error. The symbol should be the other way i.e. > 0.05 and hence corrected.

Reviewer 2 Report

The paper entitled "Integrative approach of MAP and active antimicrobial packaging for prolonged shelf-life of composite bottle gourd milk cake" is interesting.

The work brings a very broad approach to the production of an edible biofilm, incorporated with bactericins (nisin and natamycin) and shows the result against a Gram positive bacterium and a filamentous fungus. It also presents the sensorial profile of the packaged product, using this biofilm and the increase in its shelf life.

However, some questions need to be answered.

1) why the authors used Bacillus cereus as a microbiological model and Aspergillus niger?

2) why wasn't a gram negative bacterium also evaluated?

3) how was the contamination of the petri dishes with Bacillus and Aspergillus carried out? what is the cell concentration and how was it quantified?

4) in the shelf life test of the food, was it contaminated with bacteria and fungi, or was the food in its standard conditions?

5) What does SPC (line 261) mean?

6) Figure 3.1b needs to contain the letters for interpretation.

Author Response

First and foremost, we are thankful for the reviewer for critical review of the manuscript so that the gaps could be filled meticulously.

  1. The choice of microbiological organisms for the present study was based on previous literature surveys. However, in addition to this, Bacillus cereus has also been treated as a representative gram-positive bacteria commonly prevalent in most Indian sweets, because of abundant moisture and nutrient-dense food whereas Aspergillus niger has its predominance occurrence owing to the surplus amount of sugars present in Indian counterpart. Most of the food science studies reflect the common predominant of the species and led to the consideration of mentioned organisms as model for the studies.
  2. The presence of gram-negative bacterium was not considered due to the vital fact of the absence of anaerobic conditions within the product which would otherwise be helpful in bringing something useful for the study. However, as a benchmark to negative bacterium presence of E. coli was found absent throughout the storage and has been marked in the script.  
  3. The agar disk-diffusion assay was employed to detect the antimicrobial action of control, nisin (NIF), and natamycin (NAF) films on two test microorganisms (Bacillus cereus and Aspergillus niger). For this, BHI and malt extract agar with two different agar concentrations, i.e., 0.75% w/v and 1.5% w/v, was prepared. To laid a layer, 20 ml hard agar was poured (1.5%) in sterile petri-plates and were allowed to solidify for 1-2 hours. Further, BHI and malt extract soft agar (0.75%) were inoculated with Bacillus cereus active culture (approximately 1x106 CFU/ml) @0.1% v/v and Aspergillus niger spore suspension (spore count adjusted to 104 to 105 CFU/ml in fresh malt extract broth) @0.1% v/v, respectively. After solidification, inoculated soft agar (5-8ml) was poured over the respective media plates and allowed to solidify.
  4. In the shelf-life test of food (storage period), the samples were not intentionally contaminated with bacteria or fungi, rather the natural microflora present in the air or commonly present in the dairy product was allowed to proliferate if it had access within the sample during the manufacturing process, which is considered as a factor of post-processing contamination.   
  5. Line 261, referring to SPC indicated the standard plate count and has been mentioned in the text. 
  6. Figure 3.1b letters have been added for interpretation

Reviewer 3 Report

The paper from Chawla and coworkers has been revised. The paper contains an interesting pieze of work about the effect of antibacterial biofilm in combination with modified atmosphera packaging using nidin and natamycin as antibiotics for prolonged self-life of bottles. The results of this study confirms the increment in the durability to 35 days of the real samples by using the integrative approach described. The results of this work has an impact in the field of nutrition and social well-being of people. The paper is well-written and deserves publication in Coating journal after minor revision has noted.

The minor points: 

-Please, clarify in the manuscript the stability of the preparate described in Figure 1. What is the the proportion of nidin and natamycin in the formulation? 

- Please, indicate in the discussion of the paper the possibility or not to extent this approach to others antimicrobials and others used.

- Please, include a Table with MIC or MIC50 values of this formulation and these values for the same proportion of antimicrobials used in this work withouth biofilms.

- More discussion are needed to justify the decrease of the Ph with time in Table 1 in the methodolgy implemented.

- Please, indicate how the parameters in Figure 5 are measured. Are these parameters of quantitative or qualitative type?

Author Response

First and foremost, we are thankful to the reviewer for the critical review of the manuscript so that gaps could be filled meticulously. 

  1. The whole procedure has been outlined beneath the figures. The procedure consisted of preparation of the stock solution and further making up the desired concentration of antimicrobial agents’ i.e. 5000 IU for nisin and 30 µg of natamycin in solution, which has been detailed in the caption placing an asterisk on the figure.
  2. As per the kind suggestion of the reviewer, a few lines regarding the possibility of other antimicrobial have been added on page 7 line 256 onwards. Hope this will suffice the objective of the correction asked.  
  3. The MIC values tested for the formulation from the stock solution have been detailed and depicted in figure 1, considering (potency ≥ 900 IU/mg) & (1000µg/ml), for nisin and natamycin, respectively. Henceforth, a similar procedure could be opted for varying the concentration of antimicrobials without biofilm.   
  4. Effect of pH in relation to time has been elaboratively undertaken. 
  5. The parameters of figure 5 were measured employing a 9-point hedonic scale. Samples were provided to the panellists and qualitatively the changes in sensory parameters were assessed based on the perception of sensory parameters.  

Round 2

Reviewer 2 Report

After responding to the adjustments made, the material is ready for publication